# The Memory-CD8+-T-Cell Response to Conserved Influenza Virus Epitopes in Mice Is Not Influenced by Time Since Previous Infection

**DOI:** 10.3390/vaccines12040419

**Published:** 2024-04-15

**Authors:** Josien Lanfermeijer, Koen van de Ven, Marion Hendriks, Harry van Dijken, Stefanie Lenz, Martijn Vos, José A. M. Borghans, Debbie van Baarle, Jørgen de Jonge

**Affiliations:** 1Center for Infectious Disease Control, National Institute for Public Health and the Environment, 3720 BA Bilthoven, The Netherlands; 2Center for Translational Immunology, University Medical Center Utrecht, 3584 CX Utrecht, The Netherlands; 3AstraZeneca, 2594 AV Den Haag, The Netherlands; 4DICA (Dutch Institute for Clinical Auditing), 2333 AA Leiden, The Netherlands; 5Deventer Ziekenhuis, 7416 SE Deventer, The Netherlands; 6MSD Animal Health, 5830 AA Boxmeer, The Netherlands; 7Virology & Immunology Research, Department Medical Microbiology and Infection Prevention, University Medical Center Groningen, 9700 RB Groningen, The Netherlands

**Keywords:** Influenza virus, vaccine, T-cell response, aging, conserved epitopes, booster, mice

## Abstract

To protect older adults against influenza A virus (IAV) infection, innovative strategies are imperative to overcome the decrease in protective immune response with age. One approach involves the boosting of CD8+ T cells at middle age that were previously induced by natural infection. At this stage, the immune system is still fit. Given the high conservation of T-cell epitopes within internal viral proteins, such a response may confer lasting protection against evolving influenza strains at older age, also reducing the high number of influenza immunizations currently required. However, at the time of vaccination, some individuals may have been more recently exposed to IAV than others, which could affect the T-cell response. We therefore investigated the fundamental principle of how the interval between the last infection and booster immunization during middle age influences the CD8+ T-cell response. To model this, female mice were infected at either 6 or 9 months of age and subsequently received a heterosubtypic infection booster at middle age (12 months). Before the booster infection, 6-month-primed mice displayed lower IAV-specific CD8+ T-cell responses in the spleen and lung than 9-month-primed mice. Both groups were better protected against the subsequent heterosubtypic booster infection compared to naïve mice. Notably, despite the different CD8+ T-cell levels between the 6-month- and 9-month-primed mice, we observed comparable responses after booster infection, based on IFNγ responses, and IAV-specific T-cell frequencies and repertoire diversity. Lung-derived CD8+ T cells of 6- and 9-month-primed mice expressed similar levels of tissue-resident memory-T-cell markers 30 days post booster infection. These data suggest that the IAV-specific CD8+ T-cell response after boosting is not influenced by the time post priming.

## 1. Introduction

Influenza A virus (IAV) infection leads to higher morbidity and mortality with increasing age [1]. Therefore, yearly vaccination is used as a strategy to protect the older population against IAV infection. This has reduced the severity of disease and lowered the hospitalization rate amongst older adults [1,2]. However, the efficacy with which immune responses are induced by IAV vaccination also decreases with age as a result of the aging immune system, and vaccine failure occurs in a significant part of the older population [3]. Therefore, novel strategies are needed to improve the protection of the older population. One strategy would be to vaccinate earlier in life, at middle age, when individuals are still able to mount a strong and long-lasting immune response to vaccination. 

Unfortunately, for the majority of the current IAV vaccines—which primarily induce humoral responses against the globular domain of the surface proteins hemagglutinin (HA) and neuraminidase (NA)—this strategy is not suitable. Due to mutations in these domains over time (antigenic drift), the virus tends to escape from the vaccine-induced antibody response [4,5]. The antibody response that would be induced by vaccination at middle age is thus unlikely to still mediate sufficient protection against circulating IAV strains at an older age.

In contrast to most HA- and NA-specific antibodies, T cells can recognize conserved epitopes of IAV, which can respond to variants within or even across influenza virus subtypes. Pre-existing IAV-specific T-cell responses to conserved proteins are associated with decreased severity of disease in the absence of neutralizing antibodies [6,7,8]. Moreover, it has been suggested that in older adults, the T-cell response against IAV is a better correlate of protection than the antibody response [9]. Despite the high circulation of influenza viruses in the population, the T-cell response to internal conserved influenza proteins is very heterogenous, rendering a part of the population less protected [7,10]. Thus, here lies the opportunity for vaccines that target conserved internal proteins to induce a durable broadly reactive T-cell response. These could either be live attenuated influenza vaccines [11,12], which are directly applicable, or novel, so-called universal influenza vaccines that specifically target internal proteins and may be more potent [13,14].

While T cells are thus interesting vaccine targets, the T-cell compartment unfortunately undergoes significant changes with aging, leading to a decreased response. In older adults, memory T cells tend to have impaired proliferation and lytic capacities [15], IAV-specific T-cell numbers are lower, and dominant public T-cell clones are observed less frequently [16,17]. Primary IAV infection in old mice has been shown to lead to lower IAV-specific T-cell responses and a less diverse antigen-specific T-cell receptor (TCR) repertoire compared to IAV infection in young mice [18]. It may therefore be beneficial to boost the T-cell response at middle age to enhance protection against IAV at an older age. In line with this, Valkenburg et al. showed that the height of the IAV-specific T-cell response and its TCR repertoire diversity in old mice (22 months of age) could be retained at the same level as that in young mice if mice were primed with IAV at a very early age (6 weeks old) [19]. The boosting of the T-cell response might also benefit the maintenance of tissue-resident-memory T (T_RM_) cells in the lung [20,21]. Lung T_RM_ cells play a critical role in the protection against heterologous influenza infections, and the loss of this population leads to reduced protection against heterosubtypic influenza infection in mice [21,22]. Additionally, increasing age and repeated exposure to influenza can impact the formation and survival of T_RM_ cells in the lung [20,23]. 

Thus, boosting the influenza T-cell response at middle age by vaccines that target conserved internal proteins could potentially increase protection against IAV infection at an older age. However, the high level of heterogeneity in infection history in the human population may influence the vaccine response. One aspect of this heterogeneity is the time since last infection, which raises the question of whether an influenza-vaccine-induced T-cell response is negatively affected when the time between vaccination and the previous infection is increased. 

We set out to investigate the fundamental principle of how the time between antigen exposures influences the IAV-specific T-cell response at middle age using mouse-adapted IAV strains. Mice were primed with IAV at either 6 or 9 months of age and received a heterosubtypic IAV booster infection at middle age, i.e., at 12 months of age. We examined the T-cell response in the spleen, lung, and bone marrow, as well as the antigen-specific TCR repertoire after IAV infection. Just before the booster, mice that were primed at a younger age (6 m primed mice) showed lower IAV-specific T-cell responses in spleen and lung than mice primed at a later age (9 m primed mice). Only in the BM, the IAV-specific T-cell responses was comparable in early- and late-primed mice. The differences in the T-cell response in the spleen and lung before the booster did not result in an altered memory-T-cell response after the booster infection in terms of T-cell frequencies, repertoire diversity, and the expression of T_RM_ markers. These findings suggest that the boosting of a pre-existing T-cell response in middle-aged mice is not affected by the time since last infection. 

## 2. Materials and Methods 

### 2.1. Animals

A population of 5.5-month-old female C57BL/6J (black/6) mice (Janvier Labs) arrived at the Animal Research Centre (Bilthoven, The Netherlands) 2 weeks before start of the study for acclimatization. Mice were inspected daily and were provided food and water ad libitum. Mice were euthanized prior to scheduled termination if they reached the humane endpoints, which were defined as >20% weight loss, inactivity, pumping breath, bulging, feeling cold, and ruffled coat. If mice showed >20% weight loss but none of the other humane endpoints, mice were not sacrificed. None of the mice reached a humane endpoint during the study. However, one mouse was found dead the day after infection, and one during the study (unrelated to infection). For (scheduled) termination, mice were anesthetized with an isoflurane/oxygen mixture and bled via orbital puncture. Influenza infections were likewise performed under anesthesia with isoflurane/oxygen mixture to minimize suffering.

### 2.2. Viruses

Influenza A/PR/8/34 virus (PR8; NIBSC code 16/108) and H7N9 reassortant influenza virus based on the A/PR/8/34 backbone (H7N9/PR8, NIBRG-268) were obtained from the National Institute for Biological Standards and Control (NIBSC, Hertfortshire, UK). Both these viruses consist of the same internal proteins and M2, but differ only in their HA and NA surface proteins. The influenza viruses were grown on MDCK cells in MEM medium (Gibco; Thermo Fisher Scientific) supplemented with 40 µg/mL gentamicin, 2 µg/mL TPCK-treated trypsin, and 0.01 M Tricine (all from Sigma-Aldrich). At >90% cytopathic effect (CPE), the suspension was collected and spun down (4000× *g* for 10 min) to remove cell debris. Supernatant was collected, aliquoted, and subsequently frozen at −80 °C. The 50% tissue-culture infectious dose (TCID_50_) was determined via serial dilution in octuplicate on MDCK cells. After 6 days of culturing, cytopathic effect (CPE) was scored and TCID_50_ values were calculated using the Reed and Muench method.

### 2.3. Study Design

Based on data gathered from a pilot experiment, a power analysis suggested a minimal group size of 8 mice to find a 1% difference in tetramer-specific CD8+ T-cell frequencies with a power of 0.8 between groups. Due to the relatively long duration of the study and the risk of age-related deaths, we increased the group size to 10 mice. The placebo group consisted of 6 mice. For practical reasons, the experiment was divided into two parts, each with 5 (experimental groups) or 3 (placebo group) mice per subgroup. Mice were randomly distributed over the groups and housed per subgroup in filtertop Macrolon III cages and accommodated with cage enrichment (igloos and nestlets). All data shown are thus from two independent experiments (per time point).

Mice were infected with 10^2^ TCID_50_ H7N9/PR8 influenza virus intra-nasally (50 µL) at 6 months of age (6 m primed) or at 9 months of age (9 m primed) while the rest of the mice received PBS. At 12 months of age, all mice, except for the placebo group, were infected with 10^3^ TCID_50_ H1N1 influenza A/PR/8/34 virus (PR8). Groups that did not receive a virus infection at 6, 9, or 12 months of age instead received PBS intra-nasally at these time points. The response against PR8 was measured before the booster (day 0), 10 days post booster (10 dpb), and 30 days post booster (30 dpb). 

### 2.4. Sample Collection

During dissection, heparin blood, spleen, lungs, and bone marrow (BM) were collected and used on the same day for measurements.

Blood: Erythrocytes were lysed using ACK buffer (KHCO_3_ 0.01 M, NH_4_Cl 0.15 M, Na_2_EDTA 0.1 mM).Spleen: Spleens were homogenized, passed through 70 µm filters (BD biosciences), and washed with RPMI 1640 containing 10% FCS, 100 U/mL penicillin, streptomycin, and glutamate (P/S/G). Erythrocytes were lysed using ACK buffer.Lung: After collection, lungs were minced into 1 mm sized chunks using scissors and incubated in 3 mL 2.4 mg/mL collagenase A (Roche) and 1 mg/mL DNAse (Roche) in RPMI1640 suspension for 30 min at 37 °C. Subsequently, the tissue suspension was diluted with 7 mL washing medium (RPMI1640 + 1%FCS + 2 mM EDTA + 1× P/S/G) and mashed over a 70 µm cell strainer using a plunger. The resulting suspension was centrifuged for 5′ at 500× *g* to remove the collagenase, after which erythrocytes were shocked using ACK buffer. The cells were then washed, transferred over a 70 µm cell strainer, and resuspended in 1 mL of stimulation medium (RPMI1640 + 10% FCS + 1× P/S/G) for ELISpot, cytokine-FACS, and dextramer staining.BM: Before cutting the femurs at both ends, muscles and residue tissues surrounding the femur were removed. A 25-gauge needle and 10 cc syringe filled with ice-cold RPMI (10 mL) were used to flush the bone marrow out of the femur into a 70 µm nylon cell strainer placed in a 50 mL Falcon conical tube. Next, the tissue was smashed through the cell strainer and washed with medium (RPMI1640 + 10%FCS + 2 mM EDTA + 1× P/S/G).

### 2.5. ELISpot

Pre-coated mouse IFNγ–ELISpot (ALP) plates (Mabtech) were used according to the manufacturer’s protocol. Cells of the spleen, lung, and BM were stimulated with 0.1 nmol/well peptide ASNENMETM (ASN, NP_366–374_ H-2 Db), SSLENFRAYV (SSL, PA_224–233_ H-2 Db), SSYRRPVGI (SSY, PB1_703–711_ H-2 Kb), or MGLIYNRM (MGL, M1_128–135_ H-2 Kb) in ELISpot plates for 20 h at 37 degrees. Controls consisted of medium, SINNFEKL (Ovalbumin _257–264_ H-2 Kb), and pma/iono stimulation. All peptides were synthesized at DGpeptides (Hangzhou city, China), with a purity of >99%. Per condition, cells were stimulated with the relevant stimuli in 125 μL stimulation medium. For each ELISpot assay, we used a high (spleen = 400,000; lung = 100,000) and low (spleen = 100,000; lung = 50,000) number of cells as we expected a large difference in responses before and after booster infection. Per time point and per tissue, either the high or low cell count condition is reported here based on which cell number yielded the fewest exceedances of the detection window (1–500 spots per well). BM ELISpot assays never reached assay saturation, and only the high condition (400,000) is reported here. After 20 h, plates were developed according to the manufacturer’s protocol. After drying the plates for 1 night, they were analyzed using the ImmunoSpot^®^ S6 CORE (CTL, Cleveland, OH, USA). Maximum count was set at 500 spots per well and responses were corrected for background responses in medium-stimulated wells.

### 2.6. Cytokine Responses Measured Using Flow Cytometry

Around 2 million lymphocytes were stimulated with ASN, SSL, or SIIN peptide for 6 h. Monensin (Biolegend) was added to the cells for the last 5 h, followed by storage o/n at 4 °C. The next day, cells were washed twice with FACS buffer (2 mM EDTA, 0.5% BSA in PBS) and extracellular staining was performed for 30 min at 4 °C in 100 µL FACS buffer with the following antibody mix: CD44(IM7)-BrilliantBlue515, CD103(2E7)-BrilliantBlue700, CD4(RM4-5)-AF700, Fixable Viability Stain 780, CD8a(53-6.7)-BrilliantViolet510, CD62L(MEL-14)-BrilliantViolet650, CD49a(Ha31/8)-BrilliantViolet786, CD69(H1.2F3)-PE/Cy7, KLRG-1(2F1)-PE-CF594, CD3e(145-2C11)-Brilliant UV395 (all BD Biosciences), and CD127-(A7R34)-BV711 (Biolegend). After washing, cells were fixated and permeabilized with BD Cytofix/Cytopmerm kit (BD Biosciences) according to the manufacturer’s protocol. Cells were then stained intracellularly with IFNγ(XMG1.2)-APC, IL-2(RMP1-30)-BrilliantViolet421 and TNF(MP6-XT22)-PE (all BD) for 20 min at 4 °C. After washing twice, each pellet was resuspended in FACS buffer and measured on an LSR Fortessa X-20 (BD). Data were analyzed using FlowJoTM Software v10.6.2 (BD). IFNγ responses were not corrected for medium background responses.

### 2.7. Antigen-Specific CD8+ T Cells Measured Using Flow Cytometry

Approximately 2 million splenocytes, 2 million lung-derived lymphocytes, 4 million BM cells, and lysed whole blood were used for dextramer staining. Cells were stained using the commercial H-2 Db ASNENMETM-APC and H-2 Db SSLENFRAYV-PE (Immudex, Virum, Denmark) for 20 min at RT in the dark. Surface staining was added containing the following antibodies: CD44(IM7)-BrilliantBlue515, CD103(2E7)-BrilliantBlue700, CD4(RM4-5)-AF700, Fixable Viability Stain 780, CD8a(53-6.7)-BrilliantViolet510, CD62L(MEL-14)-BrilliantViolet650, CD49a(Ha31/8)-BrilliantViolet786, CD69(H1.2F3)-PE/Cy7, KLRG-1(2F1)-PE-CF594, CD3e(145-2C11)-Brilliant UV395 (all BD Biosciences), and CD127-A7R34-BV711 (Biolegend). These were incubated for 30 min at 4 degrees. After washing twice, cells were resuspended in FACS buffer. Acquisition was performed on an LSR Fortessa X-20 (BD), and data analyses were performed using FlowJo v10.6.2 (BD).

### 2.8. UMAP

FlowSOM and UMAP analyses were performed in R [24] using the CATALYST package [25]. Prior to clustering, the data were pre-processed in FlowJoTM Software v10.6.2 (BD) by first gating and then exporting the CD8+dextramer+ T-cell populations. The data were then imported into R where they were transformed using logicle transformation. Next, FlowSOM analysis was performed using 10 × 10 SOMCodes merged into 12 metaclusters. FCS files containing less than 50 cells were not included in the analysis. Finally, UMAP analysis was performed. UMAP plots were colored using FlowSOM metacluster IDs.

### 2.9. Isolation of Dextramer-Specific T Cells for T-Cell-Receptor Analyses

CD8+ T cells were isolated from PBMCs using a negative-selection microbeads kit (Miltenyi Biotec) according to the manufacturer’s protocol. Next, CD8+ T cells were labeled at room temperature for 20 min with H-2 Db ASNENMETM-APC and H-2 Db SSLENFRAYV-PE (Immudex, Virum, Denmark). Next, surface staining was performed using the following mAbs: CD3(17A2)FITC, CD4(GK1.5)- BrilliantViolet510, and CD8(53.6.7)-BrilliantViolet786 (all BD). CD3+CD4−CD8+dextramer+ cells were then sorted directly into RNAlater (Ambion Inc. Applied Biosystems) using a FACS Melody (BD) and subsequently stored at −80 °C for TCRβ clonotype analysis.

### 2.10. Preparing TCRβ cDNA Libraries for Sequencing

mRNA was isolated with the RNA microkit (Qiagen) according to the manufacturer’s protocol. Next, the 5′ RACE-based SMARTer Mouse TCR α/β profiling kit (Takara Bio, San Jose, CA, USA, Inc.) was used following the manufacturer’s protocol but only using the TCRβ-specific primers to perform TCR sequencing. The PCR products were cleaned up with AMPURE XP clean-up beads (BD) and sequenced via Illumina MiSeq paired-end 2 × 300 nucleotide sequencing.

### 2.11. TCRβ Clonotype Analysis

First, demultiplexed samples were merged using tool Paired-End read merger [26]. Next, the c lonotype information was identified by aligning the sequences to reference TRBV and TRBJ genes from the raw sequence data using RTCR [27]. Clonotypes were defined by their CDR3 amino acid sequences.

To clean the data from possible errors and contamination, TCR sequences were only accepted when they consisted of at least 100 sequencing reads. This threshold was determined based on different cutoffs to make sure it would not lead to a qualitative bias in our results. Diversity was calculated using the Simpson’s diversity index value [28]. The Simpson’s index value ranges between 0 and 1, with 0 representing minimal diversity and 1 representing maximal diversity. Richness was calculated as the number of distinct TCR sequences in equally sized subsamples through iterative sampling (100,000 times) without replacement of 100,000 reads.

### 2.12. Statistics

Groups were compared using a 1-way or 2-way ANOVA, after which results were corrected for multiple testing using the two-stage linear step-up procedure of Benjamini, Krieger, and Yekutieli with a false discovery rate of 10% [29]. Multiple testing correction was performed per assay, tissue, and day (e.g., IFNγ–ELISpot, lung, day 0). In cases where a 1-way or 2-way ANOVA could not be performed, groups were compared using the Mann–Whitney U test. For all analyses, *p*-values < 0.05 (after correction for multiple testing) were considered statistically significant. Data were analyzed using GraphPad Prism 9.1.0.

## 3. Results

### 3.1. Study Design

To study the effect of time since last infection on boosting the pre-existing IAV-specific CD8+ T-cell response in middle-aged mice, we primed mice with an H7N9 influenza A reassortant virus (H7N9/PR8) based on the backbone of the H1N1 influenza A/PR/8/34 virus either at 6 months (6 m primed mice) or at 9 months (9 m primed mice) of age. At this age, mice are mature and have a fully developed immune system [30]. To assure a homologous boost of the T-cell response to the internal proteins and no interference of previously induced neutralizing antibodies, we subsequently boosted these mice with H1N1 influenza A/PR/8/34 virus (PR8) at 12 months of age (0 days post booster (dpb), Figure 1A). The age of 12 months was chosen to resemble middle-aged adults [30,31], the target group of early influenza vaccination. We additionally included a group of naive mice that only received the PR8 infection at 12 months of age (previously uninfected mice). Immune responses were investigated prior to the booster (day 0) and 10 and 30 days dpb with PR8.

### 3.2. Lower Baseline T-Cell Response in 6 m Primed Mice Compared to 9 m Primed Mice

We compared responses in the spleens and lungs of 6 m primed and 9 m primed mice at day 0 to investigate whether time since last infection influenced the T-cell response before booster infection. Lung-derived lymphocytes and splenocytes were stimulated in an IFNγ–ELISpot assay with four different conserved influenza-specific CD8+ T-cell epitopes from different proteins with differences in immunodominance: ASNENMETM (ASN, NP_366–374_ H-2 Db), SSLENFRAYV (SSL, PA_244–253_ H-2 Db), SSYRRPVGI (SSY, PB1_703–711_ H-2 Kb) and MGLIYNRM (MGL, M1_128–135_ H-2 Kb), ranked from highest to lowest level of immunodominance. Lung T-cell responses (measured using IFNγ–ELISpot assay) against SSL (*p* = 0.0159) and SSY (*p* = 0.0025), but not ASN and MGL, were significantly higher in the 9 m primed mice compared to the 6 m primed mice (Figure 1B). In the spleen, we detected significantly higher responses in 9 m primed mice against ASN (*p* = 0.0437) and SSL (*p* = 0.0437), with a similar trend towards higher responses in 9 m primed mice for SSY (*p* = 0.0614) (Figure 1C). Correspondingly, the sum of the individual peptide-specific T-cell responses was lower in the 6 m primed mice than in the 9 m primed mice in both lung (*p* = 0.0227) and spleen (*p* = 0.0753; Figure 1D,E) samples. 

We also measured frequencies of CD8+ T cells, recognizing the two most immunodominant epitopes, ASN and SSL, through dextramer staining in lung, spleen, and blood samples (Appendix A). In line with the IFNγ–ELISpot results, SSL-specific CD8+ T-cell frequencies were lower in the lungs of 6 m primed mice compared to those of the 9 m primed mice (*p* = 0.0439) while in the spleen, both ASN-specific (*p* = 0.0197) and SSL-specific (*p* = 0.0264) T-cell frequencies were lower in 6 m primed mice (Figure 1F). The blood showed the same trend as observed in the spleen although the differences were not significant (ASN *p* = 0.0753, SSL *p* = 0.0563). Together, these data suggest that the IAV-specific T-cell response at 12 months of age is lower in 6 m primed mice compared to 9 m primed mice although the significance of the observed differences seems dependent on the epitope and compartment investigated. 

Next, we investigated the phenotype of ASN-specific and SSL-specific T-cells in blood, spleen, and lung samples at baseline before the second infection (i.e., day 0). We found that the fraction of central-memory (CM) T cells (CD62L+, CD44+) within the ASN-specific and SSL-specific CD8+ T-cell populations was significantly higher in 6 m primed mice compared to 9 m primed mice (Appendix A). Correspondingly, the fraction of cells in the effector memory (EM) population (CD62L−, CD44+) was lower in 6 m primed mice compared to 9 m primed mice. We also measured the expression of PD-1 as this inhibitory marker has been suggested to play a role in memory formation and to restrain the early expansion of virus-specific CD8+ T cells during acute respiratory infections [32]. The percentage of PD-1+ T cells within the ASN-specific T-cell population of 9 m primed mice was significantly higher than in 6 m primed mice (Appendix A). No such significant difference was observed for the SSL-specific T-cell population. In conclusion, not only the magnitude but also the phenotype of the antigen-specific memory-T-cell population changes over time, with a shift towards a central memory phenotype occurring longer after priming.

### 3.3. Similar Differences in T-Cell Responses to Booster Infection in 6 m and 9 m Primed Mice

Next, we investigated whether the differences that we observed between the 6 m primed and 9 m primed mice would affect the recall response against PR8 infection at 12 months of age. In addition, we compared this response to a de novo response in 12-month-old mice that had not been infected before (previously uninfected). Infection led to a decrease in weight for all mice (Figure 2A). However, mice that were previously uninfected showed a significantly higher weight loss (up to 24% at day 9) compared to the previously infected (both 6 m primed and 9 m primed) mice. This shows that the prior infection with H7N9 provided protection against disease upon heterosubtypic infection with PR8. The 6 m primed and 9 m primed mice showed similar kinetics with only a significant difference on day 11 after infection. 

We investigated the T-cell response at 10 and 30 dpb to cover both the effector phase as well as the early memory response [33]. In most cases, the cellular immune response against the four peptides (ASN, SSL, SSY, and MGL) at 10 and 30 dpb in both the lung and spleen did not differ between 6 m primed and 9 m primed mice, as determined via IFNγ–ELISpot (Figure 2B,C). There were a few exceptions: the responses against SSY at 30 dpb in the lung (*p* = 0.0048) and spleen (*p* = 0.0100) were significantly lower in 6 m primed mice compared to 9 m primed mice. In the spleen, the MGL responses at 10 (*p* = 0.0493) and 30 dpb (*p* = 0.0133) were also lower in 6 m primed mice compared to 9 m primed mice. Strikingly, while T-cell responses against ASN, SSL and SSY clearly increased after booster infection in the lung and spleen in 6 m primed and 9 m primed mice (Appendix A), the responses against SSL, SSY, and MGL in these mice did not exceed the response in previously uninfected mice at 10 dpb (Figure 2B,C), except for the response to MGL in 9 m primed mice at 30 dpb. 

After booster infection, the IFNγ–ELISpot assays often reached the upper limit of detection, due to which we might not have been able to detect potential differences between 6 m primed and 9 m primed mice. We therefore also measured IFNγ responses using flow cytometry. IFNγ responses after the ex vivo stimulation of lung-derived lymphocytes and splenocytes with the ASN and SSL peptides showed similar responses between 6 m primed and 9 m primed mice at both 10 and 30 dpb (Figure 3A,B). When comparing previously uninfected mice with primed mice, results were in line with the IFNγ–ELISpot results. ASN-specific responses in lung and spleen were clearly higher in 6 m primed and 9 m primed mice compared to previously uninfected mice while SSL responses did not significantly differ between primed and previously uninfected mice (Figure 3A,B). 

We also analyzed the antigen-specific CD8+ T-cell frequencies through flow cytometry using dextramer staining. In agreement with the IFNγ–ELISpot results, SSL-specific T-cell frequencies in the lungs and spleen at 10 and 30 dpb did not differ significantly between 6 m primed and 9 m primed mice (Figure 3C,D). In contrast, ASN-specific T-cell frequencies were higher in the lungs (*p* = 0.0473) and spleen (*p* = 0.0490) in 6 m primed mice at 10 dpb. The variation within groups was large, however, with percentages ranging from 0 to 45% in the lung at 10 dpb. ASN-specific T-cell frequencies were clearly boosted in 6 m primed and 9 m primed mice compared to previously uninfected mice at 10 dpb. SSL-specific T-cell frequencies, in contrast, did not exceed the response in previously uninfected mice at 10 and 30 dpb. Despite our finding that the phenotype of IAV-specific T-cells before booster infection differed significantly between 9 m primed and 6 m primed mice, we did not find any significant differences in the percentages of EM and CM cells between the two groups during the recall response (Appendix A). 

In conclusion, the higher T-cell response we observed in 9 m primed mice compared to 6 m primed mice before the booster (day 0) only marginally affected the T-cell response 10 and 30 days after the booster infection. The differences between 6 m primed and 9 m primed mice that we detected through IFNγ–ELISpot assay were observed for the sub-dominant SSY and MGL peptides. Responses against ASN were clearly boosted in 6 m primed and 9 m primed mice as they exceeded the responses of previously uninfected mice in both the spleen and lung. In contrast, we did not find significant differences for SSL-specific responses between the three treatment groups. 

### 3.4. IAV-Specific T-Cell Responses Are Similarly Maintained in the Bone Marrow in 6 m Primed and 9 m Primed Mice

The bone marrow (BM) is generally regarded as a reservoir for long-term immunological memory, and it has been shown that antigen-specific T-cell populations stay relatively constant in BM while they decrease in the spleen [34]. We wondered to what extent the IAV-specific T-cell response in the BM in 6 m primed and 9 m primed mice had been maintained before the recall response against PR8. Despite the observed differences in the baseline T-cell response against the four influenza epitopes in the spleen and lung (Figure 1B–F), the baseline responses in the BM were comparable between 6 m primed and 9 m primed mice (Figure 4A,B). This suggests that the IAV-specific T-cell response in the BM is more stable over time than the responses in spleen and lung, supporting the hypothesis that the BM is a reservoir for long-term T-cell memory. After the booster infection, T-cell responses in the BM against ASN, SSL, and SSY increased similarly in 6 m primed and 9 m primed mice and stayed stable until 30 dpb while MGL responses did not significantly increase after the booster (Appendix A). 

### 3.5. Lung-Derived T Cells of 6 m Primed and 9 m Primed Mice Express Similar Levels of T_RM_ Markers after Booster

Tissue-resident-memory T (T_RM_) cells in the lung have been shown to be important in protection against heterosubtypic influenza infections [21,22]. T_RM_ cells are usually classified based on the expression of CD69, CD103, and/or CD49a. Since the antigen-specific T_RM_-population in the lung is known to decrease with time, we wondered whether 6 m primed mice showed reduced expression levels of these markers. In order to establish whether this analysis was feasible, we first investigated whether CD69, CD103, and CD49a expression levels differed between the lung and blood. As we measured several other markers for T-cell functioning (e.g., PD-1, KLRG-1) in combination with the expression of T_RM_ markers CD49a, CD69, and CD103, a multitude of subpopulations could be identified. In order to reduce the complexity and improve visualization, we performed uniform manifold approximation and projection (UMAP) analysis with the flow cytometry data of SSL-specific and ASN-specific T cells. An additional benefit of UMAP is that it does not look at expression in a binary manner (e.g., positive vs negative populations), enabling us to compare different gradations of expression. For this UMAP, we combined the flow cytometry data from ASN-specific and SSL-specific CD8+ T cells of 6 m primed and 9 m primed mice before the booster (day 0, Appendix A). We separated the T-cell population into 12 distinct clusters and compared the prevalence of each cluster between tissues (blood vs lung) and antigen specificity (ASN vs SSL; Figure 5A, Appendix A). When comparing the UMAP results between the blood and lung, it is clear that clusters 1, 2, 3, and 4 are almost exclusively present in the lung. Importantly, cluster 1 is marked by high CD103 expression while clusters 2, 3, and 4 are characterized by high CD69 expression. Cluster 3 is additionally characterized by elevated CD49a expression. The higher prevalence of clusters 1–4 in the lung compared to the blood was significant for the SSL-specific population (Appendix A). For ASN-specific T cells, only cluster 2 was significantly increased in the lung (*p* = 0.0228). 

After we identified that populations marked by the expression of CD49a, CD69, and CD103 were elevated in the lung, we wanted to compare the antigen-specific CD8+ T-cell population from 6 m primed mice with 9 m primed mice using a more traditional binary approach. We determined which part of the ASN- and SSL-specific T-cell population was expressing CD49a, CD69, and/or CD103 (Figure 5B) and compared this between tissues (blood vs lung) and infection histories (6 m primed vs 9 m primed mice). In agreement with the UMAP, a significantly higher proportion of ASN-specific and SSL-specific T cells expressed CD69 or CD103 in the lung compared to the blood (Appendix A). However, despite previous reports that CD49a expression is associated with T_RM_ cell retention in the lung [35,36], we could not find higher CD49a expression in T cells derived from the lung. We did find that CD49a expression in general was high on ASN-specific and SSL-specific T cells, reflecting that these cells had been recently activated (Appendix A). Differences between 6 m primed and 9 m primed mice were limited to CD103, with CD103 expression levels being significantly reduced in 6 m primed mice (*p* = 0.0044 for ASN, *p* = 0.0009 for SSL). The proportion of antigen-specific T cells that expressed both CD69 and CD103—a combination that is typically used to identify T_RM_ cells—was likewise higher in the lungs of 9 m primed mice. 

Next, we assessed how the expression of CD49a, CD69, and CD103 changed after a second influenza infection. We performed another UMAP analysis in which we included lung-derived ASN-specific and SSL-specific T cells from previously uninfected mice and 6 m primed and 9 m primed mice 30 days after PR8 infection (Appendix A). The sizes of most clusters did not differ between 6 m primed and 9 m primed mice; only cluster 1 was slightly less pronounced in 6 m primed mice. Correspondingly, we did not find significant differences in the proportion of T cells expressing CD49a, CD69, CD103, or both CD69 and CD103 between these groups. Compared to 6 m primed and 9 m primed mice, a larger proportion of lung-derived SSL-specific T cells of previously uninfected mice expressed CD49a, CD69, CD103, and a combination of CD69 and CD103 at 30 dpb (Figure 5C). In contrast, the expressions of CD49a, CD69, and CD103 on lung-derived ASN-specific T cells from previously uninfected mice were not significantly different compared to those on ASN-specific T cells from primed mice. In conclusion, at 30 dpb, the expression of T_RM_ markers on lung-derived T cells in previously uninfected mice was at least as high as, or even higher than, in 6 m primed or 9 m primed mice. Additionally, while CD103 expression was initially lower on lung-derived T cells of 6 m primed mice compared to 9 m primed mice, this difference disappeared after a second influenza infection.

As T_RM_ cells are potent IFNγ producers [37,38,39], we wondered whether the expression of T_RM_ markers was related to cytokine expression. To study this, we reanalyzed the flow-cytometry data from Figure 3A, which depicts IFNγ responses within CD8+ T cells at 10 and 30 dpb. We determined the percentage of cells expressing IFNγ within populations that were either negative or positive for CD49a, CD69, or CD103 and calculated the ratio of IFNγ-expressing cells in the subset that was positive versus the subset that was negative for a certain T_RM_ marker (e.g., ratio = %IFNγ+ within CD49a+/%IFNγ+ within CD49a−). Cells that expressed CD49a and CD69 were more likely to also express IFNγ on both day 0 and 30 dpb (Figure 5D). This was especially true for CD69: cells were 10 times more likely to produce IFNγ if they expressed CD69. CD103 expression was associated with higher IFNγ responses in the ASN-specific population on day 0, but not for SSL-specific CD8+ T cells or at 30 dpb. These findings indicate that expression of T_RM_ markers—at least to some extent—correlates with increased IFNγ expression. We did not observe any differences in the frequency of IFNγ-expressing T_RM_ cells between 6 m primed and 9 m primed mice 30 dpb. 

### 3.6. IAV-Specific TCR Repertoire Diversity Is Maintained between 3 and 6 Months after Infection

In older adults, a loss of dominant public clones against IAV has previously been observed, which might explain a decreased T-cell response to infection or vaccination at an older age [16,17]. To test whether a loss of IAV-specific clonotypes also occurs after the boosting of the antigen-specific T-cell repertoire, we sorted ASN-specific T cells from splenocytes at baseline and 10 dpb and analyzed their TCRβ sequences. TCR diversity was compared by calculating the Simpson’s diversity index value [28], wherein an index of 1 represents maximal diversity and an index of 0 minimal diversity. The value of Simpson’s diversity index reflects both the richness and the evenness of the TCR repertoire. Evenness is a measure of the distribution of clones in the TCR repertoire, wherein low evenness is indicative of skewing in the TCR repertoire due to the large expansion of selective clones. As a measure for evenness, we assessed the contribution of the most dominant clones by calculating the sum of the percentages of the top five largest T-cell clones to the total repertoire per sample. Repertoire richness was calculated as the number of distinct TCR sequences in equally sized subsamples to compensate for the fact that the number of reads differed between samples.

First, we compared the diversity of the ASN-specific T-cell repertoire between 6 m primed and 9 m primed mice before the booster (day 0). No significant difference in TCR diversity was observed between the two primed groups (Figure 6A, open circles). The similarity of the TCR repertoires of the 6 m primed and 9 m primed mice at day 0 was reflected in the similar levels of repertoire skewing (Figure 6B) and richness (Figure 6C) of the ASN-specific T-cell repertoire. Also, a skewing towards Vβ13-1 was observed, which was comparable for the 6 m primed and 9 m primed mice (Appendix A). We also calculated the clonal distribution of the TCR repertoire per group, per time point. This was done by ranking the top 30 clones from large to small and calculating their contributions (in frequencies) to the repertoire (Appendix A). These data suggest that despite the lower number of antigen-specific T cells at day 0 in 6 m primed mice compared to 9 m primed mice, TCR repertoire diversity is maintained from 3 until 6 months after IAV infection.

At 10 dpb, the diversity of the ASN-specific TCR repertoire in the previously uninfected group was significantly larger compared to that in the previously primed groups (*p* = 0.0067 (9 m primed), *p* = 0.0079 (6 m primed) (Figure 6A, closed circles). This was related to the significantly less skewing of the T-cell repertoire in the previously uninfected group compared to the two primed groups (*p* = 0.0173 (9 m primed) and *p* = 0.0079 (6 m primed)) (Figure 6B). There was no significant difference in Vβ13-1 segment usage between the uninfected and primed groups (Appendix A). In previously uninfected mice, the top five clones constituted ±60% of the total repertoire while the contributions of the top five clones were as high as ±75% and ±80% in 9 m primed and 6 m primed mice, respectively (Figure 6B). This suggests that the decreased diversity observed in previously primed mice was due to an expansion of the most dominant clones. This was confirmed by the clonal distribution of the TCR repertoire we estimated based on the contribution of the top 30 largest clones in frequency, ranked from largest to smallest (Figure 6D).

No significant differences in the ASN-specific TCR repertoire were observed between the 6 m primed and 9 m primed mice at 10 dpb based on repertoire diversity (Figure 6A), evenness (Figure 6B), or richness (Figure 6C). In the 6 m primed group, TCR diversity decreased significantly between baseline and 10 dpb (*p* = 0.0303), which was not observed in 9 m primed mice (Figure 6A). Together with the observation that the TCR diversity at 10 dpb was higher in the previously uninfected group than in the primed groups, suggests that boosting the antigen-specific repertoire decreases the diversity of the responding T-cell repertoire but also that time between antigen exposure has no significant impact on the responding T-cell repertoire.

## 4. Discussion

We examined the impact of the time interval between a previous infection and a booster immunization in middle-aged female mice on the IAV-specific CD8+ T-cell response. Our data show that despite the lower baseline T-cell response in 6 m primed mice compared to 9 m primed mice, the subsequent CD8+ T-cell response and repertoire diversity did not significantly differ between these groups at 10 and 30 days after booster infection. Similarly, while baseline CD103+ T_RM_ cell numbers in the lung were lower in 6 m primed mice compared to 9 m primed mice, their numbers were boosted to similar levels after reinfection. Although boosting the response to a previous influenza infection resulted in the skewing of the TCR repertoire, we observed no influence of the timing between the prime and boost on TCR repertoire diversity. This suggests that the IAV-specific CD8+ T-cell repertoire is maintained over time. Of note, since female and male mice in some instances respond differently to influenza infection, the current outcome should still be confirmed in male mice.

To study the broad IAV-specific CD8+ T-cell response, we selected four different IAV-specific epitopes that are known to induce T-cell responses in mice [40]. These epitopes are presented on either H2-Db (ASN and SSL) or H2-Kb (SSY and MGL) and differ in the level of immunodominance of the T-cell response. The response against these epitopes has already been studied extensively and is known to have different dynamics after heterologous infection [41]. Our data are in line with earlier studies in which ASN was found to induce the most immunodominant response after primary infection, followed by SSL and SSY [42], whereas MGL induces a subdominant response [43]. It has also been described that a shift in immunodominance occurs after a secondary infection, mostly because of a diminished SSL response [44], which we also observed. The dynamics of the recall responses against the IAV epitopes were independent of the time since primary infection. Only SSY responses were significantly higher in the lung and spleen in 9 m primed mice at 30 dpb as 6 m primed mice did not show an increase in the SSY response at all after the booster infection.

It was already known that the ASN-specific T-cell repertoire tends to be highly skewed towards Vβ8.3 (IMGT gene name: Vβ13-1) usage [45,46]. We observed a similar bias for Vβ13-1 across all infection groups and time points. We found that the TCR repertoire diversity during the response against the secondary infection was not influenced by the time since the last infection. The TCR repertoire diversity of the IAV-specific T-cell response decreased after boosting due to the expansion of the most dominant clones. This phenomenon has also been observed for other pathogens like listeria monocytogenes and CMV [47,48,49]. Adoptive transfer models have shown a slight narrowing of the repertoire in the recall response although this was restored during the (second) contraction phase [50]. Nevertheless, our findings seem to be in contrast with previous studies on repeated IAV infection, which found no significant differences between the T-cell repertoires after primary and secondary infection in terms of Vβ8.3 usage and showed no evidence for the emergence of new TCRβ sequences [46,51]. The same has been found for the SSL-specific T-cell response, for which the breadth of the TCR repertoire did not change upon secondary infection [52]. Also, the study conducted by Valkenburg et al., in which mice were primed at 6 weeks of age, showed that boosting at a later age did not result in changes in the ASN-specific and SSL-specific repertoire diversity [19]. This seeming discrepancy with our study may be due to differences in the methods used to analyze the TCR repertoire as (1) the high-throughput sequencing that we used leads to a higher sequencing depth and influences the estimated richness of the repertoire as more clones are detected and (2) the previously described studies investigated the diversity within the Vβ13-1 segment while we examined the whole ASN-specific T-cell repertoire. Interestingly, we also did not observe a difference in the Vβ13-1 usage after boosting. 

As the lung is the site of infection, one could question whether the TCR repertoires are comparable between the spleen and lung. Due to having a limited number of T cells from the lung, we unfortunately could not investigate the TCR repertoire of ASN-specific lung T cells. Interestingly, a previous study showed more than 70% overlap between the SSL-specific T-cell repertoires after primary IAV challenge in the lung and spleen [52]. After the secondary challenge, this overlap was even as high as 80%. This suggests that clonal frequencies measured in the spleen may constitute a good predictive read-out of the antigen-specific TCR repertoire in the lung. We generally observed similar T-cell response kinetics—in terms of IFNγ responses over time—in the lung, spleen, and blood after the booster infection. The only compartment that showed different T-cell response kinetics was the BM, which fits with previous studies showing that the number of antigen-specific T cells in the BM remains constant up to 1 year post infection [34]. Our data also suggest that antigen-specific T cells are maintained in the BM for at least 3 months as at day 0, we found comparable responses between 6 m primed and 9 m primed mice. The maintenance of the IAV-specific response in the BM could be part of the explanation why the systemic IAV-specific responses were comparable in 6 m primed and 9 m primed mice after the recall infection despite the presence of lower baseline IAV-specific T-cell levels in 6 m primed mice in other compartments. Also, we only looked at one facet of the immune response. However, for other vaccines, an analysis of the CD4+ T-cell response and/or non-neutralizing antibodies could be more relevant.

Lung T_RM_ cells contribute to protection against heterosubtypic influenza infections, and the boosting of IAV-specific lung T_RM_ cells should thus increase the level of protection [20,21,22,23,53]. Before booster infection, the expressions of the T_RM_ markers CD49a and CD69 were similar in 6 m primed and 9 m primed mice, but CD103 expression was reduced in 6 m primed mice. This suggests that the CD103+ phenotype of ASN-specific and SSL-specific CD8+ T cells declines over time and that the boosting of the T-cell response might be beneficial. Previous studies have demonstrated that CD49a is involved in the retention of T_RM_ cells in the lung tissue [35,36]. Interestingly, we found that CD49a expression was not elevated in the antigenic-specific T-cell population in the lung tissue compared to blood, even at 6 months post infection. Our finding does not necessarily contradict previous findings as CD49a might not directly mediate migration into the lungs but primarily allows T_RM_ cells to be retained after migration into the lung. It is also noteworthy that CD69 and CD103 were expressed by a larger fraction of SSL-specific compared to ASN-specific T cells. This was consistent with a previous study reporting fewer ASN-specific T cells with a T_RM_ phenotype compared to SSL-specific T cells [54]. The authors suggested that this might be due to differences in the antigen availability as (i) the presence of antigens contributes to T_RM_ cell formation [54] and (ii) influenza antigens are known to be differentially expressed over time [44].

Several studies have shown that T_RM_ cells are potent cytokine producers [37,38,39]. The expression of CD49a has been linked to the enhanced cytokine production of T_RM_ cells in the skin [55] and to a polyfunctional phenotype of lung T_RM_ cells [56]. Our finding that antigen-specific T cells expressing CD49a, CD69, or CD103 in the lung are associated with IFNγ production supports these previous reports. Lung T_RM_ cells cannot be solely identified by the expression of (a combination of) CD49a, CD69, and CD103 as some T_RM_ cells have been described to be negative for these markers [56,57]. Additionally, lymphocytes derived from the lungs comprise a mix of circulating and resident T cells, making it difficult to assess how many cells are truly lung T_RM_ cells [58]. To distinguish between lung T_RM_ cells and circulating T cells, one can perfuse the lungs to reduce contamination with circulating lymphocytes or use an intravenous (i.v.) staining method [58]. We did not perfuse the lungs for practical reasons and the i.v. staining was incompatible with the IFNγ-stimulation for flow cytometry. It is thus likely that in our study, the lung-derived lymphocytes comprised a mix of circulating and lung resident T cells. However, a significantly larger proportion of T cells derived from the lung expressed T_RM_ markers CD69 and CD103 compared to T cells derived from the blood, indicating that we did isolate a population rich in T_RM_ cells from the lungs of infected mice.

Our data show that the IAV-specific CD8+ T-cell response can be successfully boosted in middle-aged mice independent of time since previous infection. This suggests that the timing between prime and boost does not play a major role when introducing vaccination at middle age in humans although it remains challenging to translate the 3-month difference between early (6 month) and late (9 month) priming to the human situation. The turnover rate of T cells in mice is much faster compared to that in humans [59], and the process of immune aging in mature mice (>6 months) is approximately 25 times faster than in humans [30]. A period of 3 months in mice would thus translate to >6 years in humans, which is not disproportionate considering that adults are estimated to be infected with influenza every 5–10 years [60,61]. 

If one would indeed want to protect the elderly by boosting the IAV-specific CD8+ T-cell response at middle age, it is important to know how long the IAV-specific T cells can be maintained and provide sufficient protection against disease. In mice, it has been suggested that IAV-specific T cells can be maintained for a lifetime [62], which is approximately 2 years. The lifetime of the IAV-specific CD8+ T-cell response in humans remains hard to quantify. The T-cell response against the HLA-A2 restricted epitope GILGFVFTL can be detected in the majority of HLA-A2+ individuals at respectable frequencies and has been suggested to be maintained for at least 13 years [63]. However, human adults are thought to be infected with IAV every 5–10 years due to the seasonal circulation of the virus [60,64], meaning that their IAV-specific T-cell response would be boosted multiple times during life. As we found that the boosting of pre-existing IAV-specific T cells leads, amongst other things, to a decrease in the TCR diversity of the responding T cells, it would be interesting to investigate the effect of more than two repeated infections. Our data indicate that due to boosting, there is an expansion of the most dominant clones. It could be that experiencing more than two infections would lead to the exhaustion of the most dominant clones and would thereby result in the loss of dominant clonotypes, as observed in humans, or even in more skewing as has been described for CMV-specific CD8+ T cells [16,17,48]. Additionally, several reports indicate that the excessive boosting of the T-cell response might have adverse effects. Studies using other pathogens or vaccinations have shown that boosting CD8+ T cells repeatedly can result in a decrease in the cytolytic potential, cytokine production, and proliferative capacity [65,66,67] and repertoire diversity of the responding T cells [47,48,49]. Further research is needed to unravel the exact effect of taking multiple boosts on the IAV-specific T-cell response. 

Taken together, we found that the recall CD8+ T-cell responses to IAV did not significantly differ in early- and late-primed middle-aged mice. This suggests that the T-cell response after vaccination with a T-cell-inducing influenza vaccine in humans may not be affected by the time since previous IAV infection. Although our study only covers one aspect of the heterogeneity between humans—the timing between two infections/vaccinations—this is promising for the development and testing of new IAV vaccines that focus on inducing cellular responses in the middle-aged population to provide protection at an older age.

## Figures and Tables

**Figure 1 vaccines-12-00419-f001:**
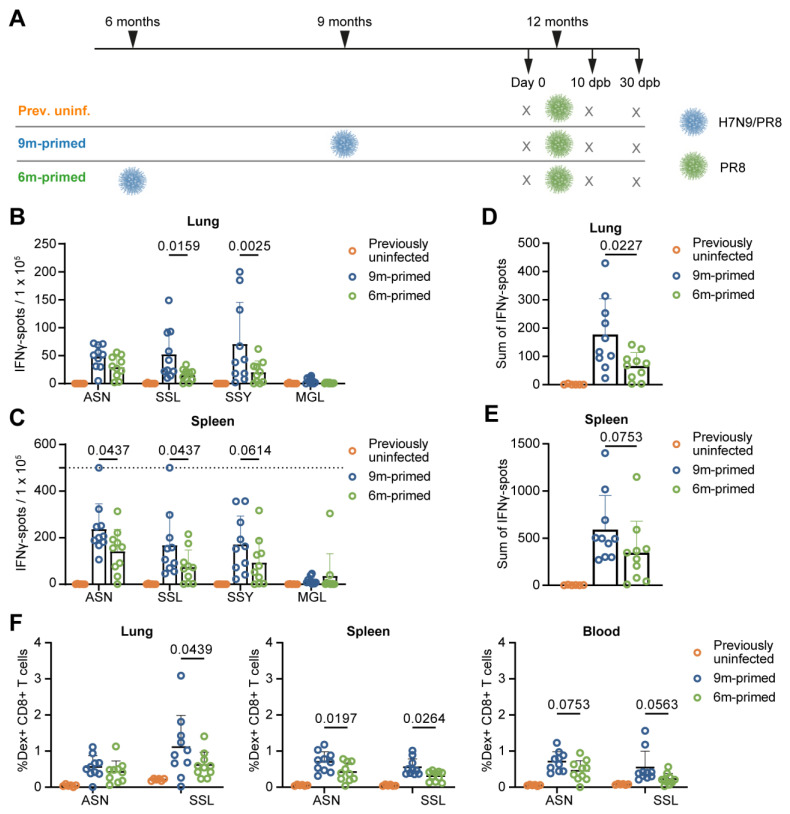
Lower baseline T-cell response in 6 m primed mice compared to 9 m primed mice. (**A**) Study layout depicting the prime-boost strategy. Mice were infected with H7N9/PR8 at 9 (9 m primed) or 6 (6 m primed) months of age. Mice not infected with H7N9/PR8 received a mock infection with PBS instead (prev. uninf.). All groups received a booster infection with PR8 at 12 months of age. Mice were sacrificed at day 0 (before booster) and 10 and 30 days post booster (dpb). (**B**–**E**) Cellular responses in IFNγ–ELISpot of lung-derived lymphocytes (**B**,**D**) and splenocytes (**C**,**E**) after restimulation with the IAV-specific epitopes ASN, SSL, SSY, and MGL at day 0. The IFNγ response is shown per peptide stimulation (**B**,**C**) or as a sum of the total response (**D**,**E**). ELISpot responses have been corrected for background (minus medium stimulation). No responses against a negative-control OVA peptide were detected. The horizontal dotted line depicts the upper limit of detection of the assay. (**F**) ASN-specific and SSL-specific T-cell frequencies measured in lung, spleen, and blood samples at day 0 with use of dextramers and depicted as frequency of total CD8+ T cells. (**B**–**F**) Results are depicted as individual mice (open circles) with means and standard deviations. For statistical testing, only 6 m primed and 9 m primed groups were compared. All data shown are from two independent experiments (per time point).

**Figure 2 vaccines-12-00419-f002:**
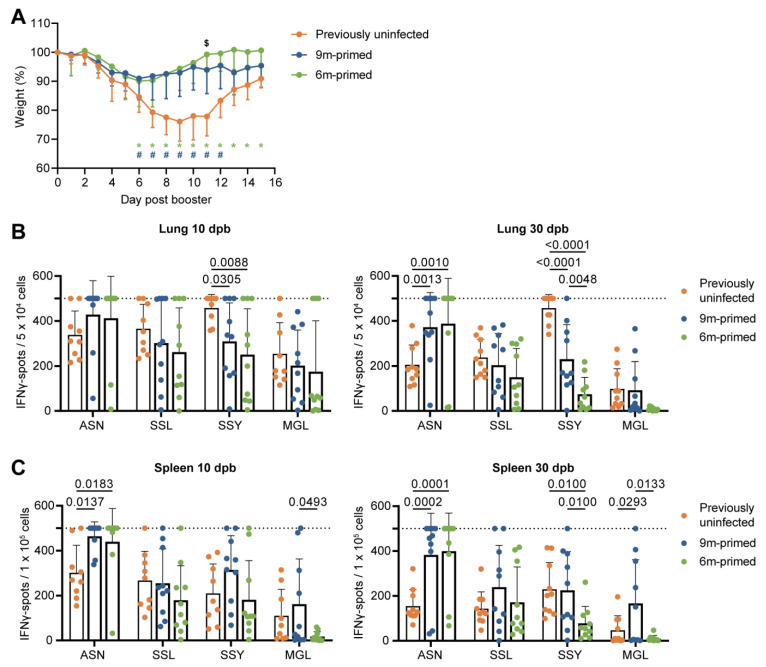
Differences before booster do not lead to differences in IFNγ response against a new IAV infection. (**A**) Body weights of mice infected with PR8 at day 0 post booster. Symbols in the graph depict significant differences between groups (* = 6 m primed vs. previously uninfected; # = 9 m primed vs. previously uninfected; $ = 6 m primed vs. 9 m primed). (**B**,**C**) Cellular responses in IFNγ–ELISpot assay of lung-derived lymphocytes (**B**) and splenocytes (**C**) after restimulation with IAV-specific epitopes at 10 and 30 days post booster (dpd). ELISpot responses have been corrected for background (minus medium stimulation). No responses against a negative-control OVA peptide were detected. Horizontal dotted lines depict the upper limit of detection of the assay. Results are depicted as individual mice (filled circles) with means and standard deviations. All data shown are from two independent experiments (per time point).

**Figure 3 vaccines-12-00419-f003:**
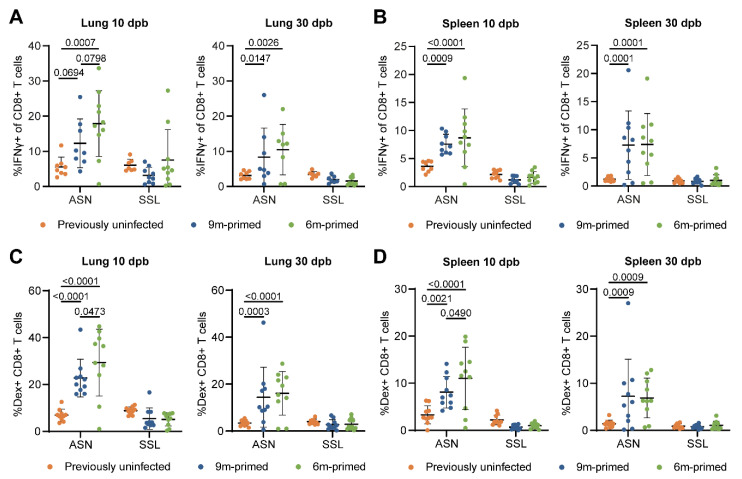
Differences before booster do not lead to differences in IFNγ-expressing and Ag-specific T-cell frequencies against a new IAV infection. (**A**,**B**) IFNγ-positive CD8+ T cells after stimulation with the ASN or SSL peptide in lung (**C**) and spleen (**D**) at 10 and 30 dpb. (**C**,**D**) ASN- and SSL-specific T-cell frequencies were measured in lung (**A**) and spleen (**B**) with dextramers and depicted as frequencies of total CD8+ T cells at 10 and 30 days post booster (dpb). (**A**–**D**) Results are depicted as individual mice (filled circles) with means and standard deviations. All data shown are from two independent experiments (per time point).

**Figure 4 vaccines-12-00419-f004:**
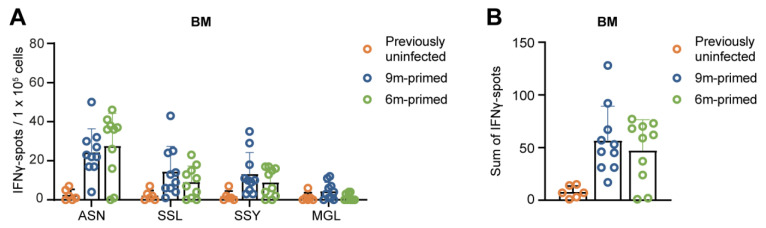
IAV-specific T-cell responses are similarly maintained in the bone marrow in 6 m primed and 9 m primed mice before booster. (**A**,**B**) IFNγ responses against four IAV-specific epitopes in bone marrow (BM) in 9 m primed and 6 m primed mice before booster as measured using IFNγ–ELISpot assay. The IFNγ response is shown per single peptide stimulation (**A**) or as a sum of the total response (**B**). ELISpot responses are corrected for background (minus medium stimulation). No responses against a negative-control OVA peptide were detected. Results are depicted as individual mice (open circles) with means and standard deviations. For statistical testing, only 6 m primed and 9 m primed groups were compared. All data shown are from two independent experiments (per time point).

**Figure 5 vaccines-12-00419-f005:**
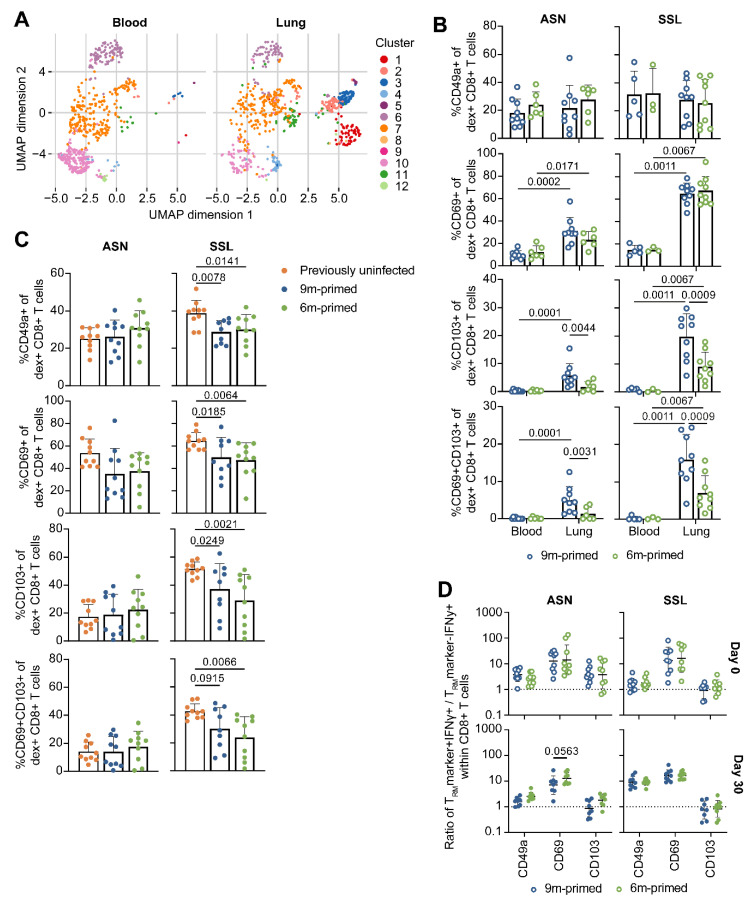
Expression of T_RM_ markers initially differs between 6 m primed and 9 m primed groups, but is similar after booster**.** (**A**) Uniform manifold approximation and projection (UMAP) analysis to compare blood- and lung-derived ASN-specific and SSL-specific CD8+ T cells of 6 m primed and 9 m primed mice before booster (day 0). Groups and antigen-specific CD8+ T cells derived from 6 m primed and 9 m primed mice were combined and analyzed separately for blood and lung-derived CD8+ T cells. (**B**) Frequency of ASN-specific and SSL-specific CD8+ T cells at day 0 expressing the T_RM_ marker CD49a, CD69, CD103, or both CD69 and CD103. (**C**) Frequency of ASN-specific and SSL-specific CD8+ T cells within lung lymphocytes of 30 dpb expressing the T_RM_ marker CD49a, CD69, CD103, or both CD69 and CD103. (**D**) IFNγ production after stimulation with ASN or SSL peptide within CD8+ T cells that are negative or positive for the T_RM_ marker CD49a, CD69, or CD103 at 0 and 30 dpb. Results are depicted as a ratio (T_RM_ marker+ IFNγ+/T_RM_ marker- IFNγ+), in which a ratio > 1 means that cells positive for a certain T_RM_ marker are more likely to express IFNγ than cells not expressing that T_RM_ marker. For (**A**–**D**), samples containing less than 50 ASN-specific or SSL-specific cells were excluded from analysis. Results are depicted as individual mice (open circles = before booster, closed circles = after booster) with (**B**,**C**) means and standard deviations or (**D**) geometric means and geometric standard deviations. All data shown are from two independent experiments (per time point).

**Figure 6 vaccines-12-00419-f006:**
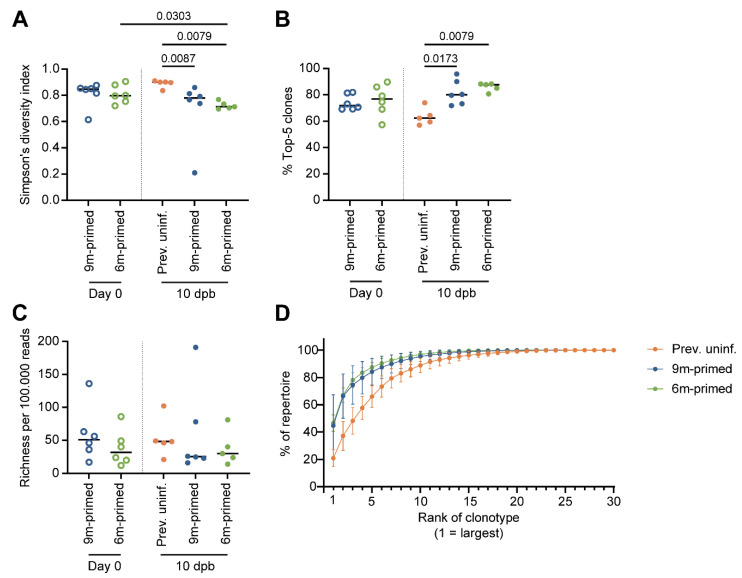
ASN-specific TCR repertoire diversity after boosting is not influenced by time between antigen exposure. (**A**) Repertoire diversity of the ASN-specific T-cell repertoire, calculated with the Simpson’s diversity index value before booster (day 0, open circles) and 10 days post booster (dpb) (filled circles). (**B**) Evenness of the ASN-specific T-cell repertoire, calculated as the contribution of the top 5 largest clones per sample in percentages. (**C**) Richness of the ASN-specific T-cell repertoire, calculated through iterative sampling (100,000 times), normalized to 100,000 reads per sample. (**D**) Clonal distribution within the TCR repertoire, ranking the top 30 most prevalent clones starting from the largest clone, plotted as the cumulative frequency at 10 dpb. In (**A**–**C**), results are depicted as individual mice (open circles = before booster, closed circles = after booster, separated by a dotted line) with means. In (**D**), results are depicted as averages per group (closed circles) with standard deviations. All data shown are from two independent experiments (per time point).

## Data Availability

All raw data files are being stored in-house on backed-up servers and are available upon reasonable request made to the corresponding author. Correspondence and requests for materials should be addressed to J.d.J.

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
