# Peer review of "The Memory-CD8+-T-Cell Response to Conserved Influenza Virus Epitopes in Mice Is Not Influenced by Time Since Previous Infection"

_vaccines, 2024, doi:10.3390/vaccines12040419_

Round 1
Reviewer 1 Report (Previous Reviewer 1)
Comments and Suggestions for Authors
The authors have sufficiently addressed the previous concerns raised by the Reviewer.
Author Response
We again thank the reviewer for the energy and time to critically review our manuscript allowing us to improve.
Reviewer 2 Report (Previous Reviewer 2)
Comments and Suggestions for Authors
Review of the manuscript “The Memory CD8+ T-Cell Response to conserved Influenza virus epitopes in Mice is Not Influenced by Time Since Previous Infection”.
The manuscript present many improvements since its first revision, nevertheless in order to be published some issues must be considered or changed on the current version.
The authors previously addressed the reviewer’s criticisms in the following manner:
1) The Title should clearly state that this response is to conserved viral antigens, and that this study was conducted exclusively in female mice, that as we all known, are far more susceptible to influenza (IAV) infection in general (they lose more weight during infection) than males. The other alternative (if you don’t want to modify the title) would be to perform a small experiment with males, or maybe, somebody else already did it). The title should be something like this after modifications:
The memory CD8+ T-Cell response to viral conserved proteins is not influenced by time since previous infection in female mice.
We changed the title from:
The Memory CD8+ T-Cell Response in Mice is Not Influenced by Time Since Previous Infection
To:
The Memory CD8+ T-Cell Response to conserved Influenza virus epitopes in mice is not influenced by time since previous infection.
However, we did not add that the mice were female. Although we agree that there is a difference between the immune response against influenza virus infection between male and female mice, this is of minor importance to the message and should therefore not be covered by the title. As also covered by comment 9), we have strict ethical rules in the Netherlands on how many animals to use, therefore only one gender was used, as using both males and females would have meant we had to use twice as many animals.
Reviewer last comment: if the tittle must not be changed, at least it should be acknowledged that the different response to influenza infection in male vs female mice could lead to different results than the obtained here for females (maybe in males -where the illness is not that severe- there is a difference in the response due to the time of immunization…). Or al least, it could be an outlook of the current work, but somewhere in the work must be stated (being the Discussion the proper place for this).
6) A whole protein control is also missing in your priming experiments, could you justify why you did not used it?
We are not sure what the reviewer means with this remark. If the reviewer means to immunize mice as a control in the first time infection, we are not sure what this would control for. A viral infection is the most potent way to induce a cellular response and protein vaccination is not and requires adjuvants. Or is it meant to serve as a control in the ELISPOT? In that case we do not see the relevance as peptide pools are optimal stimulators and we have included relevant negative and positive controls.
Reviewer last comment: Yes, of course I meant ELISPOT and IFN-γ-secretion by FC. Ionomycin/PMA and unrelated peptide are mandatory controls, these are not under discussion here. I am pointing that when you incubated your cells from different organs, you could give a whole protein also, instead of just peptides. This will give you a hint whether the antigen processing and presentation is also the same or different between treatments, and the activation achieved (secretion of IFN-γ, etc, -whichever is your readout-) will be reflecting a state more closely related to “physiological” levels of presentation (which could not be the case of your nanomolar peptides binding every MHC-I in the neighborhood). Nevertheless, it is not mandatory, but is a good control to have.
8)Since weigh is really the only “clinical” feature that is measured during IAV infection you should move the panel A from supplementary materials Figure 3, to the main body of the paper and discuss it on the results more profoundly. You can choose to what figure you add it (could be figure 3 or 4 for instance). I also recommend to have an extra panel where you compare the performance of the different immunization / treatments during the days of maximal weight loss (days 9, 10 and 11 -I guess-).
We agree with the reviewer that Panel A of Sup. Fig. 3 should be moved to the main figures. We incorporated this in figure 2 (new panel A) as this is the first figure that presents data of the second infection. We did not see the benefit of adding an additional panel to compare the maximum weight loss on specific days. Since the 6m-primed and 9m-primed mice experience a very different infection compared to the previously uninfected groups, the days of maximum weight loss also differ significantly between groups. This would make a comparison rather difficult to interpret.
Reviewer last comment: Good. Days of maximal weight loss could be different, the thing to compare is weight loss, not on which day it happened. You still could do it if days between treatments are different (as expected), just take 48 or 72 h of maximal loss and plot the average. Almost effortless right?
You do not need to change thing in the manuscript related to points 6) and 8), just add a brief statement about what I suggest at the end of 1).
Best regards,
The reviewer.
Author Response
We have included a statement on female vs male mice in the discussion and added female a few times to make more clear we used female mice. We added the following
line 29: female
line 212: female
line 621: Off note, since female and male mice in some instances respond differently to an influenza infection, the current outcome should still be confirmed in male mice.
We thank the reviewer for explaining the other comments and for the time and energy to critically review the manuscript so we could improve.
This manuscript is a resubmission of an earlier submission. The following is a list of the peer review reports and author responses from that submission.
Round 1
Reviewer 1 Report
Comments and Suggestions for Authors
In the manuscript submitted by Lanfermeijer et al, the authors attempt to determine if T-cell protective immune responses vary depending on initial exposure to the T-cell epitopes from influenza A virus (IAV) lose efficacy by middle age (following a re-infection of IAV). The Authors performed a number of experiments in mice, based on T-cell induction at 6- and 9-months of age. The Authors looked at several parameters, including the analysis of CD8+ T-cells in the spleen and lung, prior and following an additional exposure of IAV at middle age (i.e. 12 months). An analysis of cell markers associated with memory tissue-resident T-cells, as well as the TCR repertoire was also studied. In essence, while some difference could be observed between the 6-month and 9-month groups prior to boosting, the differences become more limited following IAV boosting, suggesting that T-cell priming is not significantly affected over time when if an individual is re-exposed at middle age.
This manuscript is well written; and considering the amount of information, the figures are well-presented and explained. The Authors also come to a logical conclusion based on the data that they presented. The Reviewer believes that the manuscript is suitable for publication, provided that the following concerns are addressed:
Major Concerns:
1) The Reviewer does not understand why so many figures are presented as Supplemental Figures. Generally speaking, SFs are meant to show the results of experiments that are less consequential to those presented in the main manuscript. In this case, there is a significant amount of data in the SF’s which are as important (if not more) that some of the Figures presented in the main manuscript. While the Reviewer appreciates that there was a lot of data to present, splitting the data in the manner that the Authors presented made the reading of the manuscript more confusing than it had to be.
Minor Concerns:
1) Line 20—“…during live middle age”? Please correct
2) Line 90—“ad libitium” please italicize
3) Line 334—Then why is the MGL response statistically greater in 9-month 10 dpb Spleen?
4) Line 410—Why is only 6-months presented?
5) Line 416—Don’t understand why SuppFig6 and Fig 5 couldn’t be put together in one figure
6) Line 435—To the Reviewer, both groups look pretty identical in SuppFig6C
7) Line 544--But doesn't the results in lines 538-540 contradict that (i.e. 6m had less diversity compared to 9m between baseline and 10 dpb)?
Author Response
Major Concerns:
1) The Reviewer does not understand why so many figures are presented as Supplemental Figures. Generally speaking, SFs are meant to show the results of experiments that are less consequential to those presented in the main manuscript. In this case, there is a significant amount of data in the SF’s which are as important (if not more) that some of the Figures presented in the main manuscript. While the Reviewer appreciates that there was a lot of data to present, splitting the data in the manner that the Authors presented made the reading of the manuscript more confusing than it had to be.
We are unfortunately limited by the journal guidelines in the amount of main figures. Additionally, many supplemental figures contain the same data as the main manuscript, but with a different visualization. To not overburden the reader, we decided to move these figures to the Supplemental materials. We did move panel A from supplemental figure 3 to main figure 2.
Minor Concerns:
1) Line 20—“…during live middle age”? Please correct
We change the sentence.
2) Line 90—“ad libitium” please italicize
Corrected
3) Line 334—Then why is the MGL response statistically greater in 9-month 10 dpb Spleen?
We assume the reviewer refers to 30 dpb in Spleen as indeed there the response in 9-month primed mice is higher than in previously unexposed. We added ‘except for the response to MGL in 9m-primed mice at 30 dpb.’
4) Line 410—Why is only 6-months presented?
For the first UMAP, populations from 6m-primed and 9m-primed mice were combined. Based on this analysis, we can only compare blood vs lung derived T cells (Fig 5A). In the next panel (Fig. 5B), we delve into the differences between 6m-primed and 9m-primed mice. We edited the results section to clarify this.
5) Line 416—Don’t understand why SuppFig6 and Fig 5 couldn’t be put together in one figure.
Do the reviewer mean SuppFig5 and SuppFig6 or main Fig 5 and SuppFig6? The latter would be impossible as both main Fig 5 and SuppFig6 are the size of approximately a single page.
6) Line 435—To the Reviewer, both groups look pretty identical in SuppFig6C
In SuppFig6C the left and middle panel (CD49+ expression on ASN and SSL antigen specific CD8+ cells compared to the CD49+ expression in the CD8+ population as a whole.
We rewrote this section to clarify and now it reads:
‘However, despite previous reports that CD49a expression is associated with TRM cell retention in the lung [30,3135,36], we could not find higher CD49a expression in T cells derived from the lung. We did find that CD49a expression in general was high on ASN-specific and SSL-specific T cells, reflecting that these cells have been recently activated’
7) Line 544--But doesn't the results in lines 538-540 contradict that (i.e. 6m had less diversity compared to 9m between baseline and 10 dpb)?
We draw the conclusion from the data after boosting and, there we see no difference in diversity between the 6m-primed and the 9m-primed group (at 10 dpb). Indeed, the decrease in diversity in both groups decline and is significant in 6m-primed mice. The absence of a statistical significant difference in 9m-primed mice may actually be due to one outlier, which increases the variation in this group. In any case, this observation, however, does not result in a difference between 6m- and 9m-primed mice after booster.
Reviewer 2 Report
Comments and Suggestions for Authors
Review Report for authors:
The manuscript entitled “The Memory CD8+ T-Cell Response in Mice is Not Influenced by Time Since Previous Infection”, is an interesting piece of research presenting a variety of measurements of the immune response after an early (6 months) or later (9 months) influenza infection of middle age adult mice (females).
Utterly interesting is the fact that these primed populations responded similarly in almost any measured parameter to a later challenge in elderly (or pre-elderly age for mice). Nevertheless, this fine piece of research have some details to be improved previously to be in press.
Corrections:
1) The Title should clearly state that this response is to conserved viral antigens, and that this study was conducted exclusively in female mice, that as we all known, are far more susceptible to influenza (IAV) infection in general (they lose more weight during infection) than males. The other alternative (if you don’t want to modify the title) would be to perform a small experiment with males, or maybe, somebody else already did it). The title should be something like this after modifications:
The memory CD8+ T-Cell response to viral conserved proteins is not influenced by time since previous infection in female mice.
2) Line 30 to 32 of the abstract seems to be a proper place to add some word about the clinical data you have: the weigh of mice during infection or weight loss. Also (only if you want) you can add something about the “outlook” like: ‘the immune response measured here is based on the CD8 T cell, which would be protective, reducing the countless influenza immunizations and undesired side effect of current vaccinations.”
3) In Line 61, italicize et al.
4)Section 2.1 Animals: Please, include here the strain of mice used in the study and not in the Section 2.3 (design).
5)Lines 141, 145 and 150: you wrote P/S/G...I assume this is Penicillin, Streptomycin and Gentamycin, please clarify once always before using acronyms.
6)In section 2.5 you described peptides, but the whole description on which are these proteins, its functions and the argument on its importance is missing. Please define carefully the nuclear proteins of Influenza and the peptides. Are them immunodominant? subdominant? etc. Introduce your system properly please.
A whole protein control is also missing in your priming experiments, could you justify why you did not used it?
7) On line 412 you start discussing an allegedly “reduction of complexity” by using UMAP. I cannot see such a reduction and do not understand the explanatory value of such representation. May be useful when you have more than 3 dimensions, perhaps… Anyhow, FlowJo (a software that you used) has a feature to represent many populations as Gaussian bells in 3D, maybe not that fancy, but far more understandable for the reader.
Summing up, the usefulness of the UMAP in this work seems dispensable. Nevertheless, maybe somebody else can appreciate something on it that frankly I cannot. Thus, if you want lo leave it in the paper, please explain better the groups or clusters, otherwise, replace it for a 3D FlowJo graphic.
8)Since weigh is really the only “clinical” feature that is measured during IAV infection you should move the panel A from supplementary materials Figure 3, to the main body of the paper and discuss it on the results more profoundly. You can choose to what figure you add it (could be figure 3 or 4 for instance). I also recommend to have an extra panel where you compare the performance of the different immunization / treatments during the days of maximal weight loss (days 9, 10 and 11 -I guess-).
9) Regardless your power/ sample size calculation.. A good standard in science is triplicates (at least). You just did duplicates. Anyone doing experimental science in mice knows that sometimes to get smooth results take 4 or 5 repetitions of the same experiment (which also helps you with consistency). Please next time do at least, triplicates, and not duplicates.
Final comment: I wonder if it is feasible for you to discuss about the likelihood of the heterosubtipic boost affecting negatively the immune response to a pathogen, only when the time gap between infections is short, or shorter than a given time-frame (that for sure is specie-specific). One reason for this would be the immunological noise caused by the hyper variable viral surface molecules (which probably are the most antigenic too). It seems that this immune “noise” get resolved over time, since the T cell CD8+ response in many cases is stronger and the clones that will account for memory of this response get well established in the 6 month gap better than in the 3 month gap, as you showed.
Best Regards,
The reviewer.
Author Response
1) The Title should clearly state that this response is to conserved viral antigens, and that this study was conducted exclusively in female mice, that as we all known, are far more susceptible to influenza (IAV) infection in general (they lose more weight during infection) than males. The other alternative (if you don’t want to modify the title) would be to perform a small experiment with males, or maybe, somebody else already did it). The title should be something like this after modifications:
The memory CD8+ T-Cell response to viral conserved proteins is not influenced by time since previous infection in female mice.
We changed the title from:
The Memory CD8+ T-Cell Response in Mice is Not Influenced by Time Since Previous Infection
To:
The Memory CD8+ T-Cell Response to conserved Influenza virus epitopes in mice is not influenced by time since previous infection.
However, we did not add that the mice were female. Although we agree that there is a difference between the immune response against influenza virus infection between male and female mice, this is of minor importance to the message and should therefore not be covered by the title. As also covered by comment 9), we have strict ethical rules in the Netherlands on how many animals to use, therefore only one gender was used, as using both males and females would have meant we had to use twice as many animals.
2) Line 30 to 32 of the abstract seems to be a proper place to add some word about the clinical data you have: the weigh of mice during infection or weight loss. Also (only if you want) you can add something about the “outlook” like: ‘the immune response measured here is based on the CD8 T cell, which would be protective, reducing the countless influenza immunizations and undesired side effect of current vaccinations.”
We added part of the suggested sentence above in the abstract in line 23. We did not add the part of side effect, as these are not huge for influenza vaccines and it is also not the main goal of the strategy to reduce these, although it would. Due to your comment we also added “CD8+” a few times to make it more clear in the abstract that we focused on the CD8+ T cells. We also added a sentence on the protection provided by a first infection to a heterosubtypic infection.
3) In Line 61, italicize et al.
Corrected
4)Section 2.1 Animals: Please, include here the strain of mice used in the study and not in the Section 2.3 (design).
The sentence about the strain of mice has been moved from 2.3 to 2.1.
5)Lines 141, 145 and 150: you wrote P/S/G...I assume this is Penicillin, Streptomycin and Gentamycin, please clarify once always before using acronyms.
Corrected
In this case the G stands for Glutamate.
6)In section 2.5 you described peptides, but the whole description on which are these proteins, its functions and the argument on its importance is missing. Please define carefully the nuclear proteins of Influenza and the peptides. Are them immunodominant? subdominant? etc. Introduce your system properly please.
We added some more background about the conserved internal influenza virus epitopes used, however, we believe this should be mentioned in the result section and not in the methods, we added this information to line 280-281
A whole protein control is also missing in your priming experiments, could you justify why you did not used it?
We are not sure what the reviewer means with this remark. If the reviewer means to immunize mice as a control in the first time infection, we are not sure what this would control for. A viral infection is the most potent way to induce a cellular response and protein vaccination is not and requires adjuvants. Or is it meant to serve as a control in the ELISPOT? In that case we do not see the relevance as peptide pools are optimal stimulators and we have included relevant negative and positive controls.
7) On line 412 you start discussing an allegedly “reduction of complexity” by using UMAP. I cannot see such a reduction and do not understand the explanatory value of such representation. May be useful when you have more than 3 dimensions, perhaps… Anyhow, FlowJo (a software that you used) has a feature to represent many populations as Gaussian bells in 3D, maybe not that fancy, but far more understandable for the reader.
Summing up, the usefulness of the UMAP in this work seems dispensable. Nevertheless, maybe somebody else can appreciate something on it that frankly I cannot. Thus, if you want lo leave it in the paper, please explain better the groups or clusters, otherwise, replace it for a 3D FlowJo graphic.
Our decision to use UMAP for the initial analysis is because it allows us to compare T-cell populations without focusing exclusively on binary-expression patterns of two or three markers. We do however agree with the reviewer that the explanation was lacking for the readers. We rewrote section 3.5 to more clearly explain to the reader why we choose to use UMAP and how the results should be interpreted.
8)Since weigh is really the only “clinical” feature that is measured during IAV infection you should move the panel A from supplementary materials Figure 3, to the main body of the paper and discuss it on the results more profoundly. You can choose to what figure you add it (could be figure 3 or 4 for instance). I also recommend to have an extra panel where you compare the performance of the different immunization / treatments during the days of maximal weight loss (days 9, 10 and 11 -I guess-).
We agree with the reviewer that Panel A of Sup. Fig. 3 should be moved to the main figures. We incorporated this in figure 2 (new panel A) as this is the first figure that presents data of the second infection. We did not see the benefit of adding an additional panel to compare the maximum weight loss on specific days. Since the 6m-primed and 9m-primed mice experience a very different infection compared to the previously uninfected groups, the days of maximum weight loss also differ significantly between groups. This would make a comparison rather difficult to interpret.
9) Regardless your power/ sample size calculation.. A good standard in science is triplicates (at least). You just did duplicates. Anyone doing experimental science in mice knows that sometimes to get smooth results take 4 or 5 repetitions of the same experiment (which also helps you with consistency). Please next time do at least, triplicates, and not duplicates.
We agree with the reviewer that repeating experiments multiple times increases confidence. However, in the Netherlands there are strict ethical rules, which do not allow us to repeat an experiment many times. Moreover, although this is useful, it is not generally applied in science and often n=1 experiments are also published, we therefore feel that with n=2, we have done all that is within our reach and is still scientifically sound.
Final comment: I wonder if it is feasible for you to discuss about the likelihood of the heterosubtipic boost affecting negatively the immune response to a pathogen, only when the time gap between infections is short, or shorter than a given time-frame (that for sure is specie-specific). One reason for this would be the immunological noise caused by the hyper variable viral surface molecules (which probably are the most antigenic too). It seems that this immune “noise” get resolved over time, since the T cell CD8+ response in many cases is stronger and the clones that will account for memory of this response get well established in the 6 month gap better than in the 3 month gap, as you showed.
We are not entirely sure what the reviewer means, but a previous infection could also generate antibodies to HA, which also recognize conserved part of the stalk of HA. These may interfere with and limit infection. This may cause a less efficient boost as the load of the internal proteins will be less due to a reduced infection. However, to generate these antibodies requires multiple infections and in our experimental set up this would not have played a role. Moreover, we use a virus infection as a model to boost the T-cell response. In the meantime there are vaccine strategies that only target the internal proteins to avoid interference of HA-immunity.
Reviewer 3 Report
Comments and Suggestions for Authors
1. The rationale for this study, particularly in the introduction section, appears to be weak. While I acknowledge the authors' emphasis on the importance of T-cell immunity, I find the description of the current challenges associated with the influenza vaccine somewhat inaccurate. The authors do not clearly articulate the benefits of enhancing the T-cell response in middle age to strengthen protection against the influenza virus in older age. From my perspective, the need for annual revaccination arises from the virus's rapid evolution and its ability to evade previously acquired immunity rather than any shortcomings in vaccine-induced T-cell responses. It's essential to note the existence of two vaccine types— inactivated and live-attenuated—with the latter expected to elicit a superior T-cell response. Additionally, given the high prevalence of influenza, most individuals are likely to be naturally infected before reaching older adult age. Natural infection should, theoretically, result in a robust T-cell response. However, previous infections do not prevent subsequent ones. Therefore, the authors' suggestion of vaccinating individuals earlier in life when they can still mount a robust immune response to vaccination (as mentioned in lines 41-42), appears to be illogical.
2. The nature of H7N9/PR8 (NIBRG-268) is unclear. I presume it comprises six gene segments from PR8 and two gene segments encoding HA and NA of H7N9. Please clarify this explicitly to help readers understand the experimental setup.
3. The discussion appears extensive but lacks sufficient interpretation of the findings. For instance, Figures 2 and 3 illustrate that primed mice did not exhibit enhanced T-cell responses against the four peptides analyzed, except for the peptide ASN. Meanwhile, Figure 4 indicates that primed mice had higher frequencies of IFN--secreting cells against these peptides, suggesting there were memory T-cells against these peptides. Does it mean the primed mice failed to mount an anamnestic T-cell response? The discussion should delve into these observations to provide a more comprehensive interpretation of the results.
4. The result section related to Figure 5A is difficult to follow.
Author Response
- The rationale for this study, particularly in the introduction section, appears to be weak. While I acknowledge the authors' emphasis on the importance of T-cell immunity, I find the description of the current challenges associated with the influenza vaccine somewhat inaccurate. The authors do not clearly articulate the benefits of enhancing the T-cell response in middle age to strengthen protection against the influenza virus in older age. From my perspective, the need for annual revaccination arises from the virus's rapid evolution and its ability to evade previously acquired immunity rather than any shortcomings in vaccine-induced T-cell responses. It's essential to note the existence of two vaccine types— inactivated and live-attenuated—with the latter expected to elicit a superior T-cell response. Additionally, given the high prevalence of influenza, most individuals are likely to be naturally infected before reaching older adult age. Natural infection should, theoretically, result in a robust T-cell response. However, previous infections do not prevent subsequent ones. Therefore, the authors' suggestion of vaccinating individuals earlier in life when they can still mount a robust immune response to vaccination (as mentioned in lines 41-42), appears to be illogical.
We agree with the reviewer that the rationale for the study needs further clarification. Indeed, annual re-vaccination is required due to the ability of the virus to escape previously vaccine- or infection-induced humoral immunity. However, the hypothesis is that if there would be a sufficient level of T-cell memory to conserved internal proteins, the disease burden on the population level caused by the annual influenza epidemics could be reduced. We have added some text to the introduction and motivate this below:
In the first paragraph we make the case that at older age vaccines are not effective enough, and that one novel strategy is to vaccinate at middle age when the immune system is still fit. Following, we explain that the majority of current vaccines are not suitable for this strategy as these are antibody focused and the virus escapes this immunity. Then we introduce that T-cells can recognize conserved proteins of the virus and that these can reduce disease. However, despite high circulation of influenza virus, T-cell immunity to conserved proteins at the populations level is very heterogenous and is thus not sufficient. Here, vaccines that aim to induce a durable T-cell response to conserved proteins could play a role. We then explain in more detail that at older age also the T-cell response in general is less effective and therefore immunization at middle age may be an outcome. If the new vaccination strategy would be introduced, this would be done in a population with individuals with diverse influenza infection histories. We here focus on the influence of time since the last infection, which is part of this diversity, and could impact the vaccine response.
The nature of H7N9/PR8 (NIBRG-268) is unclear. I presume it comprises six gene segments from PR8 and two gene segments encoding HA and NA of H7N9. Please clarify this explicitly to help readers understand the experimental setup.
We added the following sentence to the M&M section 2.2. Viruses: ‘These viruses thus consist of the same internal proteins and M2, but differ in their HA and NA surface proteins.’
The discussion appears extensive but lacks sufficient interpretation of the findings. For instance, Figures 2 and 3 illustrate that primed mice did not exhibit enhanced T-cell responses against the four peptides analyzed, except for the peptide ASN. Meanwhile, Figure 4 indicates that primed mice had higher frequencies of IFN-ã-secreting cells against these peptides, suggesting there were memory T-cells against these peptides. Does it mean the primed mice failed to mount an anamnestic T-cell response? The discussion should delve into these observations to provide a more comprehensive interpretation of the results.
This is a known phenomenon and relates likely to the immune dominance of ASN and is found by others. We refer to this in the discussion and since others have studied this and we confirm this, we did not see the need to further discuss this and therefore only referred to other studies.
‘ These epitopes are presented on either H2-Db (ASN & SSL) or H2-Kb (SSY & MGL) and differ in the level of immunodominance of the T-cell response. The response against these epitopes has already been studied extensively and is known to have different dynamics after heterologous infection [41]. Our data are in line with earlier studies in which ASN was found to induce the most immunodominant response after primary infection, followed by SSL and SSY [42], whereas MGL induces a subdominant response [43]. It has also been described that a shift in immunodominance occurs after a secondary infection, mostly because of a diminished SSL response [44], which we also observed. ‘
The result section related to Figure 5A is difficult to follow.
We rewrote section 3.5 to more clearly explain to the reader why we choose to use UMAP and how the results should be interpreted.